# Microbiota Characterization of the Cow Mammary Gland Microenvironment and Its Association with Somatic Cell Count

**DOI:** 10.3390/vetsci10120699

**Published:** 2023-12-11

**Authors:** Jing Liu, Huan Liu, Guangjie Cao, Yifang Cui, Huanhuan Wang, Xiaojie Chen, Fei Xu, Xiubo Li

**Affiliations:** 1National Feed Drug Reference Laboratories, Feed Research Institute, Chinese Academy of Agricultural Sciences, Beijing 100081, China; 82101211277@caas.cn (J.L.);; 2Beijing Key Laboratory for Prevention and Control of Infectious Diseases in Livestock and Poultry, Institute of Animal Husbandry and Veterinary Medicine, Beijing Academy of Agricultural and Forestry Sciences, Beijing 100081, China; 3Laboratory of Quality & Safety Risk Assessment for Products on Feed-origin Risk Factor, Ministry of Agriculture and Rural Affairs, Beijing 100081, China

**Keywords:** somatic cell count, subclinical mastitis, microbiota, 16S rRNA

## Abstract

**Simple Summary:**

Subclinical mastitis is mostly caused by pathogenic infections, including bacteria, viruses, and fungi, with bacterial infections being the most serious. SCC is a relatively accurate indicator of the severity of mastitis and is widely used as a diagnostic indicator of mastitis in dairy cows. The purpose of this investigation is to explore the relationship between SCC, diversity in the microbiome, and SCM. Recent studies have suggested that SCM-related pathogens stay in isolation identification and drug resistance, and the relative lack of SCM is related to the change in microbiome diversity. Therefore, we aim to investigate the association between dysbacteriosis and the development of SCM using 16Sr RNA high-throughput sequencing and molecular diagnostic techniques to analyze the structure and abundance of microbiota in the mammary microenvironment in different SCC ranges. The roles of *Staphylococcus* spp. and *Streptococcus* spp., as well as *Enterobacter* spp., in various udder quarters and their relationship with inflammation and mastitis have been established. *Bacillus* spp. has also been associated with the development of mastitis. Among them, *Pseudomonas*, *Serratia*, and *Acinetobacter* can be used as landmark genera for the transition from the subclinical to the clinical state of bovine mastitis.

**Abstract:**

Subclinical mastitis is a common disease that threatens the welfare and health of dairy cows and causes huge economic losses. Somatic cell count (SCC) is the most suitable indirect index used to evaluate the degree of mastitis. To explore the relationship between SCC, diversity in the microbiome, and subclinical mastitis, we performed next-generation sequencing of the 16S rRNA gene of cow’s milk with different SCC ranges. The data obtained showed that the microbiota was rich and coordinated with SCC below 2 × 10^5^. SCC above 2 × 10^5^ showed a decrease in the diversity of microbial genera. When SCC was below 2 × 10^5^, the phylum Actinobacteriota accounted for the most. When SCC was between 2 × 10^5^ and 5 × 10^5^, Firmicutes accounted for the most, and when SCC exceeded 5 × 10^5^, Firmicutes and Proteobacteria accounted for the most. Pathogenic genera such as Streptococcus spp. were absent, while SCC above 2 × 10^5^ showed a decrease in the diversity of microbial genera. SCC was positively correlated with the percentage of *Romboutsia*, *Turicibacter*, and *Paeniclostridium* and negatively correlated with the percentage of *Staphylococcus*, *Psychrobacter*, *Aerococcus*, and *Streptococcus*. *Romboutsia* decreased 6.19 times after the SCC exceeded 2 × 10^5^; the SCC increased exponentially from 2 × 10^5^ to 5 × 10^5^ and above 1 × 10^6^ in *Psychrobacter*. Analysis of the microbiota of the different SCC ranges suggests that the development of mastitis may not only be a primary infection but may also be the result of dysbiosis in the mammary gland.

## 1. Introduction

Subclinical mastitis (SCM) is an inflammation resulting from infection or tissue traumatism in dairy cows, which is incidental to occur, difficult to diagnose, and hard to prevent and control [1,2], and results in low milk quality and reduced milk yield [3]. Furthermore, if not detected and treated early, SCM can turn into clinical mastitis, leading to premature culling of cows, which brings huge economic losses to the dairy farming industry [4,5].

The mammary gland microenvironment (MGME) is the environment in which the mammary epithelial cells exist, mainly including milk composition, somatic cells, and microorganisms. Somatic cells are mainly composed of macrophages, lymphocytes, neutrophils, and exfoliated mammary epithelial cells [6]. Somatic cell count (SCC) is a valid quantitative trait for the identification of the inflammation status of the mammary gland [7] and is used worldwide as an indicator of udder health in dairy animals to monitor the quality of milk indirectly [8]. Moreover, SCC in the milk of healthy cows is low, and when a cow develops a mammary infection, inflammatory factors can lead to a dramatic increase in the number of somatic cells in the milk [9,10,11]. Therefore, SCC can reflect the severity of mammary inflammation relatively accurately and play an important role in the diagnosis of SCM.

Extensive research has shown that SCM in dairy cows is mainly triggered by the invasion of pathogenic microorganisms, especially bacterial pathogens [12]. Additionally, the most frequently isolated pathogens are *Staphylococcus aureus* [13], *Escherichia coli* [14], *Klebsiella* spp., *Streptococcus* spp. [15], *Mycoplasma* spp., *Enterobacter* spp., *Bacillus* spp., and *Corynebacterium* [16]. Notably, many microorganisms colonize the mammary gland, and the invasion by other pathogenic microorganisms and the decrease in the immunity of the body cause changes in the microbiome diversity in the mammary microenvironment, and the changes in the structure and abundance of the microbiome may be closely related to the development of SCM [17]. Compared with traditional bacterial culture methods, which have the disadvantages of long detection time and few isolated bacteria, 16S rRNA high-throughput sequencing [18,19] and analysis of highly variable regions within the 16S rRNA gene can provide a relatively stable and rapid method to evaluate the abundance of the microbiome and identify pathogenic bacteria.

The purpose of this investigation is to explore the relationship between SCC, diversity in the microbiome, and SCM. A recent study suggested that SCM-related pathogens stay in isolation identification and drug resistance, and the relative lack of SCM is related to the change in microbiome diversity. Therefore, we aim to investigate the association between dysbacteriosis and the development of SCM using 16Sr RNA high-throughput sequencing and molecular diagnostic techniques to analyze the structure and abundance of microbiomes in the mammary microenvironment in different SCC ranges.

## 2. Materials and Methods

### 2.1. Samples

The studies were conducted on a dairy farm located on the outskirts of Beijing, which had a total of 832 Holstein cows. The animals received a mixed diet, were fed three times a day, and were monitored by the Afimilk^®^ health system. The cows were milked three times a day with an average milk yield of 12.2 kg, and the milking system used was DeLaval VMS™ V300. Bedding in the dairy is sand and rice hulls.

In this study, 50 lactating cows were randomly selected for milk sampling. Milk samples were taken from cows with reference to the Laboratory Handbook on Bovine Mastitis (sample collection and handling 7–12, in the USA). In brief, samples were collected at the second milking of the day, and asepsis was strictly observed during sample collection. Ethanol (70%) was used to disinfect the hands and udder teats, followed by drying the residual ethanol on the skin surface with clean paper towels before collecting milk samples. Samples of 25–30 mL of raw milk were taken at the onset of milking in 50 mL sterile plastic centrifuge tubes. Udder teats were not allowed to touch the edge of the test tubes during sampling, and the test tubes were tightly closed with sterile caps after sampling. The milk samples were stored at 4 °C immediately after collection and analyzed within 24 h.

### 2.2. SCC in the Samples

SCC was performed with a Fossmatic somatic cell detector (Foss Electric A/S, Hillerød, Denmark). The SCC DNA was dye-stained in red, and the stained somatic cells individually passed through the flow-path system, emitting red light that was amplified by the photomultiplier and finally counted by the light pulse capture. The instrument was calibrated using the Fossmatic adjustment) sample before the sample was mounted.

### 2.3. Bacterial DNA Isolation from the Milk Samples

Approximately 5 mL of milk was collected and centrifuged at 10,000× *g* for 10 min. The fat deposits were carefully wiped off with a sterile cotton swab (supernatant could be used for other experimental analysis). Microbe’s DNA sequencing was performed using the cell pellet samples, and then the samples were placed in sterile frozen tubes and stored at −80 °C.

### 2.4. DNA Extraction and PCR Amplification

Following the manufacturer’s instructions, the E.Z.N.A.^®^ soil DNA Kit (Omega Bio-Tek Inc., Norcross, GA, USA) was used to extract microbial community genomic DNA from the milk samples. Using a 1% agarose gel, the DNA extract was examined, and the NanoDrop 2000 UV-vis spectrophotometer (Thermo Scientific, Wilmington, DE, USA) was used to measure the concentration and purity of the DNA.

Primer pairs were used to amplify the bacterial 16S rRNA gene’s hypervariable region V3–V4.

The primer sequences used are as follows:

338F: 5′-ACTCCTACGGGAGGCAGCAG-3′

806R: 5′-GGACTACHVGGGTWTCTAAT-3′

An ABI GeneAmp^®^ 9700 PCR thermocycler (ABI, City of Thousand Oaks, CA, USA) was used to execute the PCR in the following mode: first denaturation for 3 minutes at 95 °C, then 27 cycles of denaturation for 30 seconds at 95 °C, annealing for 30 seconds at 55 °C, extension for 45 seconds at 72 °C, one extension for 10 minutes at 72 °C, and finalization at 10 °C.

### 2.5. Illumina MiSeq Sequencing

Majorbio Bio-Pharm Technology Co., Ltd. (Shanghai, China) followed established techniques to pool purified amplicons at equimolar concentrations and perform paired-end sequencing on an Illumina MiSeq PE300 platform/NovaSeq PE250 platform (Illumina, San Diego, CA, USA). The NCBI Sequence Read Archive (SRA) database now houses the raw reads.

In total, 606,894 high-throughput 16Sr RNA sequences were generated for 15 samples. After subsampling each sample to an equal sequencing depth (30,412 reads per sample) and clustering, 420 operational taxonomic units (OTUs) at 97% identity were obtained, with the number of OTUs ranging from 166 to 1106 per sample.

### 2.6. Statistical Analysis

The raw 16S rRNA gene sequence reads were demultiplexed, quality-filtered using fastp (version 0.20.0) [20], and merged using FLASH (version 1.2.7) [21] based on the following criteria: (i) During a 50 bp sliding window, the 300 bp reads were truncated at any site that received an average quality score of less than 20. Reads that were truncated shorter than 50 bp were discarded, along with reads that contained ambiguous characters. (ii) Only overlapping sequences longer than 10 bp were assembled. The overlap region’s maximum mismatch ratio is 0.2. Any reads that were unassemblable were thrown away. (iii) Samples were differentiated by the barcode and primers at the beginning and end of the sequence, and the sequence orientation was adjusted to allow zero mismatches for the barcode and a maximum of two mismatches for the primers.

Operational taxonomic units (OTUs) with a 97% similarity cutoff [22] were clustered using UPARSE (version 7.1), and chimeric sequences were identified and removed. The taxonomy of each OTU representative sequence was analyzed using the RDP Classifier (version 2.2) [23] against the 16S rRNA database with a confidence threshold of 0.7.

The experimental data underwent statistical analysis using the SPSS 24.0 version software.

## 3. Results

### 3.1. SCC Measurement

A total of 200 udder quarters from 50 randomly selected cows were examined, and the results are presented in Table 1. Among them, three samples were randomly selected from each of the five groups (Table 1) categorized based on somatic cell count for further analysis of microbial diversity.

### 3.2. Analysis of Bacterial Community Diversity

The diversity of microbiomes in the mammary microenvironment was higher in Groups A and B with SCC up to 2 × 10^5^ than in the other groups. When the SCC was from 2 × 10^5^ to 1 × 10^6^, i.e., (Groups C and D), the diversity of the microorganisms in the mammary microenvironment decreased significantly, while when the SCC was higher than 1 × 10^6^, i.e., Group E, the microorganism diversity in the mammary microenvironment increased again. According to the index inter-group difference test (Figure 1a) in the Shannon index of OTU, there were significant and highly significant differences between A and C, A and D, and B and C, and there were no significant differences between the other groups.

Pan and Core OTUs are used to characterize changes in total and core species as the sample size increases. Pan OTUs refer to the sum of OTUs contained in all samples to evaluate the increase in total OTUs as the sample size increases. The adequacy of the sequencing sample size was evaluated based on the degree of flattening of the pan/core species curve. As shown in Figure 1b,c, the pan/core species curve has leveled off, indicating sufficient sequencing samples.

### 3.3. Analysis of Microbial Community Composition

The most common phyla were Firmicutes, Actinobacteriota, Proteobacteria, and Bacteroidota. The values of the mean abundance of phyla for each group are presented in Table 2. The phylum Actinobacteria accounted for the most, with 42.57% and 40.78% in Groups A and B, respectively; the largest proportion of Firmicutes was 53.09% and 53.06% in Groups C and D, respectively; and the largest proportion of Firmicutes and Proteobacteria was 36.22% and 34.84%, respectively, in Group E.

The microbial community composition was analyzed at the genus level, and the 22 genera with the highest relative abundance are shown in Figure 2a. The genus level in Actinobacteriota accounted for a large proportion of *Corynebacterium*, *Romboutsia*, *Glutamicibacter*, *Dietzia*, and *Rothia*. The genus level in Firmicutes accounted for a large proportion of *Staphylococcus*, *Aerococcus*, *Streptococcus*, *Turicibacter*, *unclassified_f_Planococcaceae*, *Paeniclostridium*, *Clostridium_sensu_stricto_1*, and *Bacillus*; in the Proteobacteria phylum at the genus level accounted for a large proportion of *Psychrobacter*, *Pseudomonas*, *Escherichia-Shigella*, *Serratia*, *Acinetobacter*, *unclassified_f_Rhizobiaceae*, and the genus *Moheibacter* in the Bacteroidota phylum.

According to the heatmap of community composition at the genus level (Figure 2c), Groups A and B with SCC below 2 × 10^5^ were rich in microbiota. Pathogenic genera such as *Streptococcus* spp. were absent, while Groups C, D, and E with SCC above 2 × 10^5^ showed a decrease in the diversity of microbial genera.

The five groups of milk samples were analyzed for the significance of the differences between the dominant genera. The results (Figure 2d) showed that at the genus level, in Groups A and B with SCC less than 2 × 10^5^, we detected that the abundance of *Staphylococcus* and *Escherichia-Shigella* increased while the abundance of *Romboutsia*, *Paeniclostridium*, *Bifidobacterium* and *Solibacillus* increased.

In Groups A and B with SCC less than 2 × 10^5^ n *Romboutsia*, *Turicibacter*, *Paeniclostridium*, *Clostridium_sensu_stricto_1*, *unclassified_f_Planococcaceae*, *Truepera*, and *Moheibacter* were 5.81 to 10.4 folds higher than those in Groups C, D, and E with SCC higher than 2 × 10^5^. Similarly, *Staphylococcus*, *Psychrobacter*, *Aerococcus*, *Streptococcus*, *Glutamicibacter*, *Escherichia-Shigella*, *Pseudomonas*, *Serratia*, and *Acinetobacter* were higher in Groups C, D, and E (with SCC above 2 × 10^5^) than in Groups A and B (with SCC less than 2 × 10^5^) by a factor of 3.38–32.3, as shown in Figure 3.

## 4. Discussion

This study aimed to analyze the effect of changes in the composition of microorganisms in the mammary microenvironment with different SCCs on SCM. The microbial composition in Groups A and B with SCC below 2 × 10^5^ was compared to that in Groups C, D, and E with SCC above 2 × 10^5^. The international standard for determining mastitis infection in cows is set at 2 × 10^5^ somatic cells per milliliter of milk. If there are more than 5 × 10^5^ somatic cells per milliliter of milk, it indicates subclinical mastitis in cows. The European Union and New Zealand have set an SCC limit of less than 4 × 10^5^ per milliliter of milk in raw milk, Canada has set an SCC limit of less than 5 × 10^5^ per milliliter of milk, and the United States has set an SCC limit of less than 7.5 × 10^5^ per milliliter of milk. Furthermore, countries are considering further raising the standards for SCC to achieve even lower levels.

The pan and core species curve can combine species richness and evenness, thus reflecting the community species diversity relatively objectively [24,25]. The abundance of microbiota was higher in Groups A and B than in the other groups, and the microbiota was rich and coordinated. In Groups C and D, the SCC was between 2 × 10^5^ and 1 × 10^6^. The SCC increased, the mammary microenvironment became inflamed, and microbiota diversity decreased. In Group E, the SCC was higher than 1 × 10^6^, but microbiota diversity began to increase, probably due to the entry of multiple environmental bacteria into the mammary gland of the cow, leaving the mammary microenvironment in a state of pathological diversity equilibrium.

At the phylum level, Firmicutes increased in proportion with the increase in SCC, differentiating them from the intestinal microorganisms, and Bacteroidota was less represented in the mammary microenvironment. According to the sequencing results, SCC below 2 × 10^5^ presented a more similar colony composition. In general, in human or animal intestinal microorganisms, a high proportion of Bacteroidota is found to be the dominant group. In this study, the highest proportion of Actinobacteria was found in a healthy mammary microenvironment, and as the SCC increased, the proportion of the Firmicutes gradually increased and overtook Actinobacteria to dominate. When the SCC increased above 1 × 10^6^, Proteobacteria significantly increased from 13.54% to 34.84%. It is suggested that some genera in Proteobacteria contributed more to the development of clinical signs of mastitis in cows.

In the present study, SCC was negatively correlated with the percentage of *Romboutsia*, *Turicibacter*, *Paeniclostridium*, *Clostridium_sensu_stricto_1*, *Truepera*, and *Moheibacter*, and positively correlated with the percentage of *Staphylococcus*, *Psychrobacter*, *Aerococcus*, *Streptococcus*, *Glutamicibacter*, *Escherichia-Shigella*, *Pseudomonas*, *Serratia*, and *Acinetobacter*.

At the genus level, *Romboutsia* was created to classify the newly isolated species *Romboutsia ilealis* as well as *Romboutsia lituseburensis* [26,27]. It is noteworthy that *Romboutsia* is usually found in the human intestine in relation to the health status of a patient [19]. Some studies [28,29] comparing bacterial populations in patients with polyps or colorectal cancer with healthy mucosal samples have shown that *Romboutsia* is more abundant in healthy individuals, and others have reported that the risk of urinary tract infections in kidney transplant patients is reduced when the relative abundance of *Romboutsia* is higher. All these studies suggest that *Romboutsia* may play a key role in maintaining the health status of the host and that this genus could be a very valuable candidate species for intestinal ecological dysregulation. In our study, *Romboutsia* was reduced 6.19-fold after the SCC went over 2 × 10^5^.

*Turicibacter* consists of exclusively anaerobic, Gram-positive bacteria belonging to Firmicutes. *Turicibacter* is an anaerobic symbiont of multiple species in the upper gastrointestinal tract and is associated with bile and bile acid formation in the digestive tract [30]. In our study, after SCC exceeded 2 × 10^5^, *Turicibacter* decreased by 8-fold. In the mammary microenvironment of cows, *Turicibacter* may act as a resident microbiota; however, the exact mechanism of action is not clear [31]. According to our data, the *Turicibacter* genus increased by 8.0-fold in samples of cows with Group B compared to Group A. Recent studies based on analysis of the 16S rRNA gene and ribosomal intergenic spacer regions have demonstrated the presence of *Turicibacter* bacteria in the rumen and feces of cattle [32]. It has also been reported to exist in the intestines of pigs, rats, and insects, as well as in dairy wastewater and whole milk [33]. Due to the isolation of only one species (*Turicibacter sanguinis*), the physiological diversity of this genus is not yet well understood [34]. However, as the isolated strain is a suspected pathogen, it is possible that *Turicibacter* spp. present in the gastrointestinal tract of farm animals may cause infections or other harmful effects, which aligns with our research findings.

In this research, it was observed that there was a decrease in the presence of *Paeniclostridium* genus in cows within Groups C, D, and E with SCC above 2 × 10^5^ compared to Groups A and B with SCC below 2 × 10^5^. The representative bacteria of the genus Paeniclostridium are widespread in nature and have a wide range of physiological characteristics, as well known. Some strains of this genus are frequently found in wounds and, more often, in combination with other anaerobic and aerobic microorganisms. The pathogenicity of these bacteria in animals derives from their highly biologically active and specific β-toxins. These microorganisms are frequently isolated in diseases with symptoms characteristic of other clostridial diseases, such as sudden death in sheep, acute peritonitis in cattle and lambs, hemorrhagic enteritis, and gangrenous lesions in the genital tract of newborn cattle [35].

Thus, Paeniclostridium, which was 5.81 times higher in Groups A and B with SCC below 2 × 10^5^ than in Groups C, D, and E with SCC above 2 × 10^5^, may be present in dairy cows as a conditionally pathogenic organism that may cause acute disease when the body’s immunity is lowered.

*Staphylococcus* spp., *Streptococcus* spp., and *E. coli* spp. are the main pathogenic groups causing SCM. *S. aureus* is the predominant causative agent, often leading to more severe clinical mastitis, and coagulase-negative *Staphylococcus* also belongs to the genus *Staphylococcus*. *Streptococcus lactis*-free and *Streptococcus lactis*-stopping are the main causative agents of mastitis in cows in the genus *Streptococcus*; *E. coli* is the main causative agent of mastitis in cows in the genus *Enterobacter* [36]. In this study, *Staphylococcus* spp. increased sharply in Group C, and *Streptococcus* spp. increased sharply in Group D. *Staphylococcus* spp. and *Streptococcus* spp. are the main causative agents of bovine mastitis. Among these, *Staphylococcus aureus* is one of the most important pathogens associated with subclinical mastitis, which is persistent and easily recurrent. Our data also showed an increase in the bacterial amount of *Staphylococcus* spp. in Groups C, D, and E with SCC above 2 × 10^5^ compared to Groups A and B with SCC below 2 × 10^5^ (Figure 2d). These findings suggest that colonization of *Staphylococcus* spp. may occur in the external area of the milk ducts, leading to mammary gland stimulation, subsequent inflammatory processes, severe damage to mammary epithelial cells, and the development of subclinical mastitis.

Furthermore, infection of coagulase-negative *Staphylococcus* (CNS) has been linked to higher milk production during later lactation stages. Certain CNS pathogens, such as *Staphylococcus epidermidis* and *Staphylococcus chromogenes*, appear to exhibit greater adaptation to the mammary gland compared to others like *Staphylococcus devriesei*, *Staphylococcus xylosus*, and *Staphylococcus arlettae*, which are more commonly associated with environmental sources. These findings highlight the species- and strain-specific adaptations of *Staphylococci* [35]. In this study, *Staphylococci* were found to colonize teat apices in similar proportions among non-infected, subclinically infected, and clinically infected quarters.

However, *Psychrobacter* increased exponentially when the SCC was between 2 × 10^5^ and 5 × 10^5^ and more than 1 × 10^6^, and when the SCC was between 2 × 10^5^ and 1 × 10^6^, *Aerococcus* was 14 times higher in Groups C and D than in the other Groups. A recent study suggested that the capacity to thrive at 37 °C, which appears to be crucial for establishing colonization in the mammalian organism, is an ancestral characteristic of *Psychrobacter* that may have evolved from a pathobiont [37].

Other members of this genus include *Salmonella rubella* and *Salmonella liquefaciens*, which have also been reported to cause hospital-acquired infections. *Acinetobacter* is widely distributed in the external environment, mainly in water and soil; the bacteria of this genus are highly adherent and may be a source of reservoirs. They are also present in the skin and pharynx of healthy people, as well as in conjunctiva, saliva, gastrointestinal tract, and vaginal secretions, and are conditional pathogens. In this study, *Pseudomonas*, *Serratia*, and *Acinetobacter* were the most abundant in Group E, and their abundance was 21.9, 32.3, and 3.65 times, respectively, higher than that in the other groups. The presence of these three genera could initially predict the transformation of mastitis in cows from a subclinical to a clinical state.

## 5. Conclusions

Our study shows that SCC above 2 × 10^5^ can be used as a parameter for early warning of cryptogenic mastitis in dairy cows. The role of *Staphylococcus* spp. and *Streptococcus* spp., as well as *Enterobacter* spp., in various udder lesions and their relationship with inflammation and SCM have been established. *Bacillus* spp. has also been associated with the development of SCM. Among them, *Pseudomonas*, *Serratia*, and *Acinetobacter* can be used as landmark genera for the transition from the subclinical to the clinical state of mastitis in cows. The analysis of microbiota from different SCC ranges suggests that the development of SCM may not only be a primary infection but may also result from dysbiosis of the intramammary microbiomes.

By 16SrRNA high-throughput sequencing targeting the V3–V4 variable region of the 16S rRNA gene, we were able to analyze the microbiota present in the mammary microenvironment and identify the different bacterial genera at the time of sampling. Although this study was limited to a small number of herds, the application of 16S rRNA high-throughput sequencing holds promise for further exploring the variability of the mammary microbiota. This technique can be utilized to investigate how the microbiota composition varies in relation to udder health, cow-related characteristics, therapeutic interventions, and farm management practices.

## Figures and Tables

**Figure 1 vetsci-10-00699-f001:**
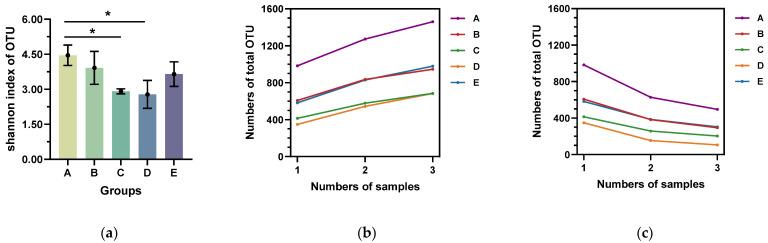
(**a**) Alpha analysis’s inter-group variability test for indices. (**b**) Pan species curve in milk samples. (**c**) Core species curve in milk samples. *, *p* value less than 0.05.

**Figure 2 vetsci-10-00699-f002:**
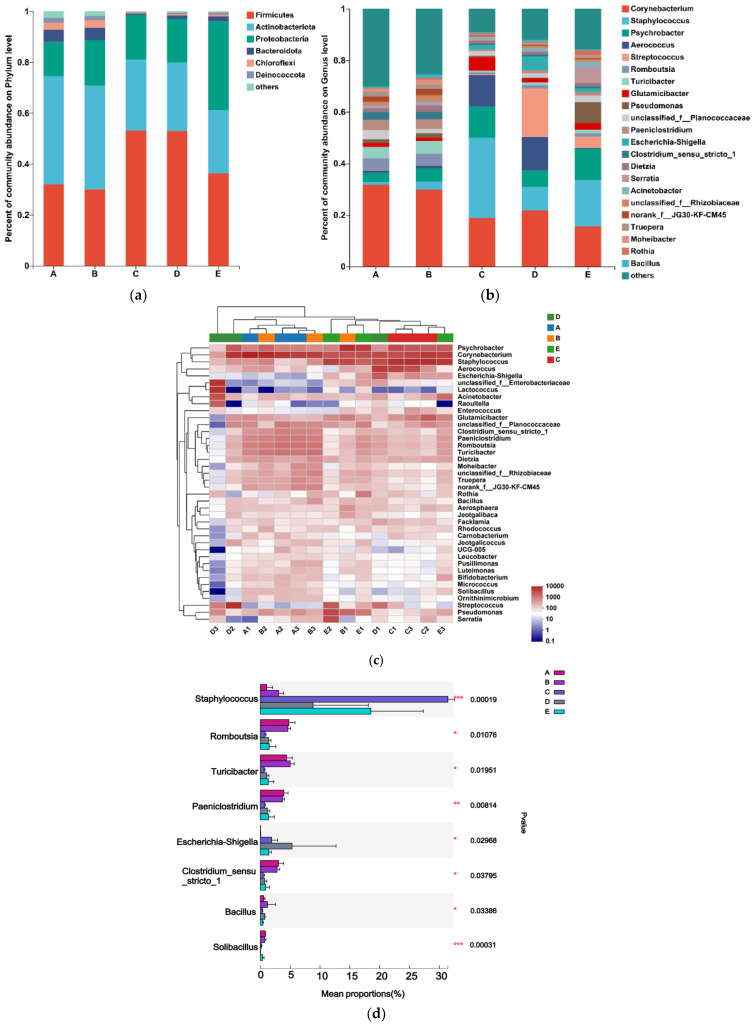
(**a**) Relative abundance of bacterial phyla in the studied groups. (**b**) Relative abundance of bacterial genera in the studied groups. (**c**) Heatmap of community composition at the genus level. (**d**) The first eight species that differed in abundance were analyzed by one-way ANOVA for abundance in five groups. *, *p* value less than 0.05; **, *p* < 0.005; ***, *p* < 0.0005.

**Figure 3 vetsci-10-00699-f003:**
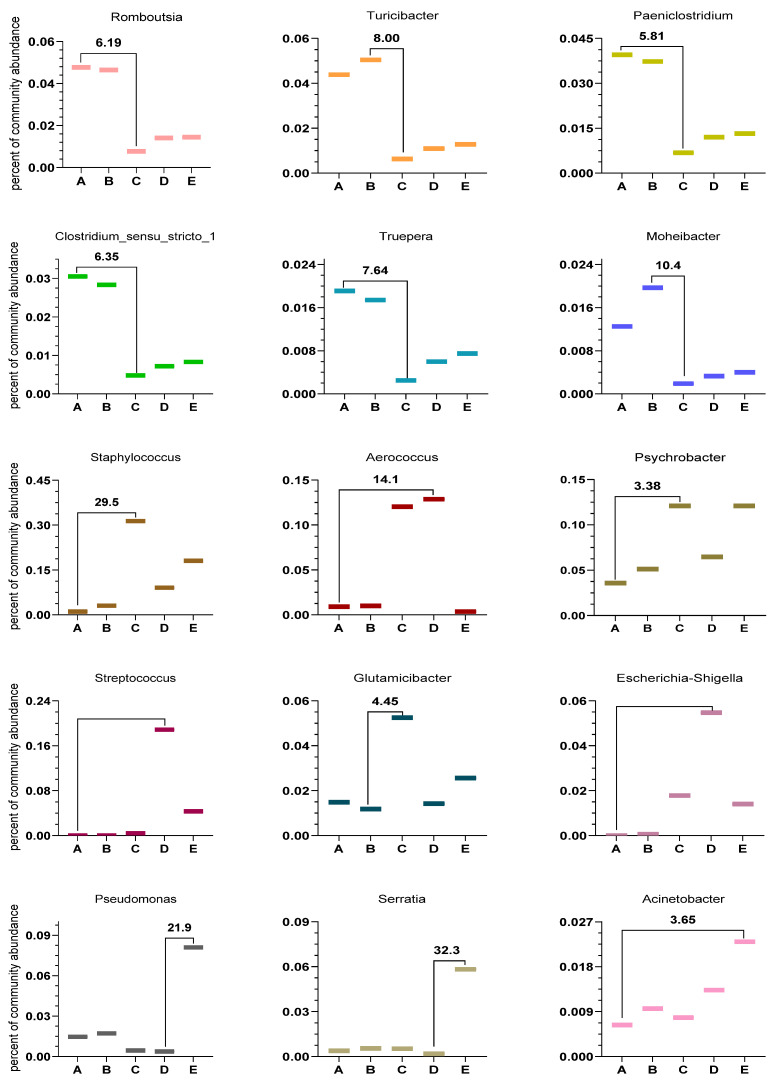
Ratio of the highest and lowest multiplicity of different species in the five groups.

**Table 1 vetsci-10-00699-t001:** SCC in the milk samples.

Indicators	Group A	Group B	Group C	Group D	Group E
Somatic cells	<1 × 10^5^	1 × 10^5^–2 × 10^5^	2 × 10^5^–5 × 10^5^	5 × 10^5^–1 × 10^6^	>1 × 10^6^
Quarters	75	73	26	15	11
proportion	37.5%	36.5%	13.0%	7.5%	5.5%

**Table 2 vetsci-10-00699-t002:** Relative abundance of bacterial phyla in the studied groups.

Mean Abundance					
Group	Firmicutes	Actinobacteriota	Proteobacteria	Bacteroidota	Others
A	0.3192	0.4257	0.1354	0.0475	0.0722
B	0.3001	0.4078	0.1797	0.0476	0.0648
C	0.5309	0.2790	0.1752	0.0066	0.0083
D	0.5306	0.2685	0.1719	0.0133	0.0157
E	0.3622	0.2500	0.3484	0.0201	0.0193

## Data Availability

Data are contained within the article.

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
