# Peer review of "Microbiota Characterization of the Cow Mammary Gland Microenvironment and Its Association with Somatic Cell Count"

_vetsci, 2023, doi:10.3390/vetsci10120699_

Round 1

Reviewer 1 Report

Comments and Suggestions for Authors

I would like to express my gratitude for offering me the opportunity to review your intriguing paper. The question of assessing dysbiosis using the well-established technique of 16S rRNA and genomic tools is not novel. However, the novel aspect, as far as my knowledge goes, is its association with the severity of inflammation.

The study is meticulously conducted and thoughtfully interpreted. Although the language is generally well-written, there are occasional issues that, even for a non-native English speaker like myself, can hinder proper comprehension. The most notable instances can be found in the attached article text.

In the discussion section, one crucial aspect have been omitted. It pertains to whether this dysbiosis is responsible for the heightened inflammation in the udder or if it is a consequence of another dysbiosis in the gut. This is significant since the interactions between gut and mammary gland microbiota are well-documented (see this paper using the same approaches: Jiang, Chenxi & Hou, Xiaolu & Gao, Xiaona & Liu, Pei & Guo, Xiaoquan & Hu, Guoliang & Li, Qingqing & Huang, Cheng & Li, Guyue & Fang, Weile & Mai, Wanrui & Wu, Cong & Xu, Zheng & Liu, Ping. (2023). The 16S rDNA high-throughput sequencing correlation analysis of milk and gut microbial communities in mastitis Holstein cows. BMC Microbiology. 23. 10.1186/s12866-023-02925-7. )

It remains unclear whether you sampled the entire mammary gland or individual quarters (to be precised). If the latter is the case, it would be easier to draw conclusions. Generally, only one quarter is contaminated, and if dysbiosis is only evident in this quarter, it could be inferred that contamination induces dysbiosis, which subsequently leads to increased inflammation. However, if you conducted whole udder sampling, you lack information regarding the differences between contaminated and healthy quarters. This makes it difficult to ascertain whether dysbiosis is a general occurrence or solely linked to infection. If it is widespread, the conclusion may differ, with dysbiosis potentially facilitating the establishment of pathogens and the development of mastitis. If a connection can be established between the same dysbiosis in the gut (or uterine flora), it could corroborate the latter explanation, which holds significant implications for veterinarians seeking to reduce mastitis in dairy cattle. This aspect must be addressed in the discussion.

please read the atached file for some details to be corrected or precised.

regards

Comments on the Quality of English Language

Author Response

Response : Thank you for the constructive suggestion.

In this study, our aim was to investigate the relationship between SCM occurrence in dairy cows and the ecological imbalance in mammary glands. To address the concern that the observed differences in SCC may be due to inflammation rather than an ecological imbalance, we designed a sequencing experiment with cows having different somatic cell counts. By analyzing the gene expression patterns and comparing them between cows with high and low somatic cell counts, we aimed to determine if the observed differences were primarily caused by inflammation or if they were indicative of an ecological imbalance in the mammary gland. The results of our experiment support the hypothesis that the ecological imbalance of mammary glands is indeed associated with the occurrence of SCM in dairy cows.

Additionally, it is important to note that further experiments and investigations are still ongoing to gain a more comprehensive understanding of the relationship between SCM occurrence and the ecological imbalance of mammary glands in dairy cows. These ongoing studies aim to explore additional factors and potential mechanisms involved in this relationship. The results from these future experiments will contribute to the growing body of knowledge in this field and may provide further insights into the prevention and management of SCM in dairy cows.

Reviewer 2 Report

Comments and Suggestions for Authors

Thank you for presenting your results on a relevant topic. I find the provided results relevant and there is probably still some way until the knowledge and understanding of the udder microbiota is fulfilled and can be applied in dairy cattle management.

I do, however, have some suggestions for improvement of the manuscript;

During the whole manuscript subclinical mastitis is mentioned. Please define what SCM is to you. There are several definitions and SCC cut-offs used over the world. I think also the introduction and aim will be more clear if you define what you mean.

Section 2.1: please add the information that you collect quarter milk samples. Also that you had criteria of no clinical signs of mastitis? What about recent antimicrobial treatment? Furthermore, I suggest, as you mention feeding and the use of afimilk, you should also add the type of barn/stalls, bedding, milking system and how many times of milking/day. Milk yield and some kind of SCC level of cows or bulk milk would also be informative. Did you consider if age or time since last antimicrobial treatment had an effect on your findings? It could be mentioned as a possibility in the discussion. Furthermore, the further description of farm and cows could be mowed to the results section.

In line 81 you mention that you use 1-5 mL for the analysis? Is that not a huge variation? I guess it will affect the sensitivity of the test. At least, if I understood correct, it should be mentioned in the discussion.

Section 2.6: significance levels used should be mentioned. Furthermore, there are no confidence intervals for your proportions and ratios – which I think is lacking. In relation to this, the relatively small sample size should be mentioned in the discussion.

Section 3.1: the information about the selection of 3 samples from each group actually should be provided in materials and methods I think. Also how these were selected (random, from same cows or?) In general the section seems not really to be formulated for the results. I suggest rewriting. What you present is the number and proportion of quarters with different levels of SCC.

line 227-228: I am not sure where this information is coming from?

line 230-236: this section is written with no reference? The statements are not the truth for all countries, therefore it should be more carefully described which study is mentioned. Furthermore, SCM is not really defined.

Conclusion: I do not understand how you come to this conclusion based on your findings. At least not how you can say anything about clinical mastitis.

Author Response

Thank you for presenting your results on a relevant topic. I find the provided results relevant and there is probably still some way until the knowledge and understanding of the udder microbiota is fulfilled and can be applied in dairy cattle management.

I do, however, have some suggestions for improvement of the manuscript;

During the whole manuscript subclinical mastitis is mentioned. Please define what SCM is to you. There are several definitions and SCC cut-offs used over the world. I think also the introduction and aim will be more clear if you define what you mean.

Response : Thank you for the constructive suggestion. We've redefined "SCM.", Subclinical mastitis (SCM) is an inflammation resulting from infection or tissue traumatisms in dairy cows see line 47.

The international standard for determining mastitis infection in cows is set at 2×105 somatic cells per milliliter of milk. If there are more than 5×105 somatic cells per milliliter of milk, it indicates subclinical mastitis in cows. The European Union and New Zealand have set a SCC limit of less than 4×105 per milliliter of milk in raw milk, Canada has set a SCC limit of less than 5×105 per milliliter of milk, and the United States has set a SCC limit of less than 7.5×105 per milliliter of milk.

Section 2.1: please add the information that you collect quarter milk samples. Also that you had criteria of no clinical signs of mastitis? What about recent antimicrobial treatment? Furthermore, I suggest, as you mention feeding and the use of afimilk, you should also add the type of barn/stalls, bedding, milking system and how many times of milking/day. Milk yield and some kind of SCC level of cows or bulk milk would also be informative. Did you consider if age or time since last antimicrobial treatment had an effect on your findings? It could be mentioned as a possibility in the discussion. Furthermore, the further description of farm and cows could be mowed to the results section.

Response: Thank you for the constructive suggestion. Added related elements to "Materials and Methods" and "Discussion", see line 86-88.

The cows were milked three times a day with an average milk yield of 12.2 kg and the milking system used was DeLaval VMS™ V300. Bedding in the dairy is sand and rice hulls.

50 lactating Holstein cows without antibiotics for 14 days were selected and 200 raw milk samples were taken.

In line 81 you mention that you use 1-5 mL for the analysis? Is that not a huge variation? I guess it will affect the sensitivity of the test. At least, if I understood correct, it should be mentioned in the discussion.

Response: Thank you for the constructive suggestion. We collected 25-30 mL of raw milk as following the method in 2.1 and then in 5 ml of raw milk was used for analysis. Corrections have been made in the article.

Section 2.6: significance levels used should be mentioned. Furthermore, there are no confidence intervals for your proportions and ratios – which I think is lacking. In relation to this, the relatively small sample size should be mentioned in the discussion.

Response: Thank you for the constructive suggestion.  We collected 25-30 mL of raw milk as following the method in 2.1 and then in 5 ml of raw milk was used for analysis. Corrections have been made in the article.

Section 3.1: the information about the selection of 3 samples from each group actually should be provided in materials and methods I think. Also how these were selected (random, from same cows or?) In general the section seems not really to be formulated for the results. I suggest rewriting. What you present is the number and proportion of quarters with different levels of SCC.

Response: Thank you for the constructive suggestion. 50 lactating Holstein cows without antibiotics for 14 days were selected and 200 raw milk samples were taken, somatic cell counts were performed on the samples, and after counting the SCC were divided into five groups, the proportions of which are shown in Table 2. We randomly selected three samples from each of the five groups.

line 227-228: I am not sure where this information is coming from?

Thank you for the constructive suggestion. Corrections have been made in the article.

Reviewer 3 Report

Comments and Suggestions for Authors

The author divided cows into different groups according to SCC in milk and analyzed the correlation between SCC and milk microbiota. The results showed that the distribution of microbiota in milk of cows with different SCC was significantly different, which proved that the occurrence of mastitis in dairy cows was closely related to mammary microbiota disturbance in addition to the infection of mammary glands by pathogenic bacteria. The results of this study are of guiding significance for the study of the pathological mechanism and prevention methods of mastitis in dairy cows.

There are still a few questions about the experiment

1.    When the SCC of dairy cows is greater than 500,000/mL, the mammary gland may have inflammatory reaction. In addition to the sequencing results, the author should test whether there is common mastitis pathogenic bacteria infection in the milk. If so, how to clarify whether the mammary gland is more susceptible to bacterial infection after the disturbance of the mammary microbiota, or whether the bacterial infection leads to the change of the inherent mammary microbiota?

2.    In lines 206-201. The authors suggested that SCC was negatively correlated with the percentage of Staphylococcus, Psychrobacter, Aerococcus, Streptococcus, Glutamicibacter, Escherichia-Shigella in the present study. It is generally believed that Escherichia coli, Staphylococcus aureus and streptococcus are common pathogens of mastitis. In this study, such pathogens were found to be negatively correlated with SCC. Why?

3.    There are some typos in the manuscript, for example: 2×105 SCC.

Author Response

  1. When the SCC of dairy cows is greater than 500,000/mL, the mammary gland may have inflammatory reaction. In addition to the sequencing results, the author should test whether there is common mastitis pathogenic bacteria infection in the milk. If so, how to clarify whether the mammary gland is more susceptible to bacterial infection after the disturbance of the mammary microbiota, or whether the bacterial infection leads to the change of the inherent mammary microbiota?

Response : Thank you for the constructive suggestion. The international standard for determining mastitis infection in cows is set at 2×105 somatic cells per milliliter of milk. If there are more than 5×105 somatic cells per milliliter of milk, it indicates subclinical mastitis in cows.

  1. In lines 206-201. The authors suggested that SCC was negatively correlated with the percentage of Staphylococcus, Psychrobacter, Aerococcus, Streptococcus, Glutamicibacter, Escherichia-Shigellain the present study. It is generally believed that Escherichia coli, Staphylococcus aureus and streptococcus are common pathogens of mastitis. In this study, such pathogens were found to be negatively correlated with SCC. Why?

Response: Thank you for the constructive suggestion. It’s mistakes. Corrections have been made in the article.” SCC was negatively correlated with the percentage of Romboutsia, Turicibacter, Paeniclostridium, Clostridium_sensu_stricto_1, Truepera, and Moheibacter, and positively correlated with the percentage of Staphylococcus, Psychrobacter, Aerococcus, Streptococcus, Glutamicibacter, Escherichia-Shigella, Pseudomonas, Serratia, and Acinetobacter.”

  1. There are some typos in the manuscript, for example: 2×105 SCC.

Response: Thank you for the constructive suggestion. Corrections have been made in the article.

Reviewer 4 Report

Comments and Suggestions for Authors

In this study, 16S rRNA sequencing technology was used to explore the characteristics of microbial flora in dairy cow mammary gland microenvironment and its relationship with somatic cell count. There are several questions in this study:

1.The final result of this study is that the occurrence of mastitis in dairy cows is related to the ecological imbalance of mammary glands. In this experiment, cows with different somatic cell count were sequenced to investigate whether it was due to inflammation, rather than an ecological imbalance in the mammary gland

2.The description in 3.1 is too similar to the method description, and the result description is not specific enough. I don't know what the outcome of this part is.

3.What is the specific meaning of mammary gland microenvironment? This has not been mentioned in previous statements and discussions.

4.The sequencing samples were milk samples, and the sequencing results could only explain the microbial differences of milk, but not the somatic cell count. Could it be proved that the microorganisms in milk could represent the microorganisms in the mammary gland microenvironment.

5.Data analysis is not mentioned in the methodology.

Comments on the Quality of English Language

Moderate editing of English language required.

Author Response

  1. The final result of this study is that the occurrence of mastitis in dairy cows is related to the ecological imbalance of mammary glands. In this experiment, cows with different somatic cell count were sequenced to investigate whether it was due to inflammation, rather than an ecological imbalance in the mammary gland

Thank you for the constructive suggestion. In this study, our aim was to investigate the relationship between SCM occurrence in dairy cows and the ecological imbalance in mammary glands. To address the concern that the observed differences in SCC may be due to inflammation rather than an ecological imbalance, we designed a sequencing experiment with cows having different somatic cell counts. By analyzing the gene expression patterns and comparing them between cows with high and low somatic cell counts, we aimed to determine if the observed differences were primarily caused by inflammation or if they were indicative of an ecological imbalance in the mammary gland. The results of our experiment support the hypothesis that the ecological imbalance of mammary glands is indeed associated with the occurrence of SCM in dairy cows.

  1. The description in 3.1 is too similar to the method description, and the result description is not specific enough. I don't know what the outcome of this part is.

Response: Thank you for the constructive suggestion. Part of result 3.1 is a simple count of somatic cell counts for all milk samples taken.

3.What is the specific meaning of mammary gland microenvironment? This has not been mentioned in previous statements and discussions.

Response: Thank you for the constructive suggestion. We've redefined " The mammary gland microenvironment", The mammary gland microenvironment (MGME) is the environment in which the mammary epithelial cells exist, mainly including milk composition, somatic cells and microorganisms, see line 52.

  1. The sequencing samples were milk samples, and the sequencing results could only explain the microbial differences of milk, but not the somatic cell count. Could it be proved that the microorganisms in milk could represent the microorganisms in the mammary gland microenvironment.

Response: Thank you for the constructive suggestion. The mammary gland microenvironment (MGME) is the environment in which the mammary epithelial cells exist, mainly including milk composition, somatic cells and microorganisms. In this research, we followed the method in 2.1 and collected raw milk samples in which the microbiota in the mammary gland microenvironment was included. Just like in gut microbiology analysis, feces are usually used as samples for the analysis of gut microorganisms.

5.Data analysis is not mentioned in the methodology

Thank you for the constructive suggestion. See line 169.
